# Performance of an Emergency Road Ambulance Service in Bhutan: Response Time, Utilization, and Outcomes

**DOI:** 10.3390/tropicalmed7060087

**Published:** 2022-05-31

**Authors:** Tshokey Tshokey, Ugyen Tshering, Karma Lhazeen, Arpine Abrahamyan, Collins Timire, Bikash Gurung, Devi Charan Subedi, Kencho Wangdi, Victor Del Rio Vilas, Rony Zachariah

**Affiliations:** 1Jigme Dorji Wangchuck National Referral Hospital (JDWNRH), Thimphu 11001, Bhutan; 2Emergency Medical Services Division (EMSD), Ministry of Health, Thimphu 11001, Bhutan; utshering@health.gov.bt; 3Department of Medical Services, Ministry of Health, Thimphu 1101, Bhutan; klhazeen@health.gov.bt; 4TB Prevention and Research Centre, Yerevan 0014, Armenia; arpine_abrahamyan@yahoo.com; 5The Union, 7500 Paris, France; collins.timire@theunion.org; 6Health Help Centre (HHC), EMSD, Ministry of Health, Thimphu 11001, Bhutan; bgurung@health.gov.bt (B.G.); dcsubedi@health.gov.bt (D.C.S.); 7WHO Country Office, Thimphu 11001, Bhutan; wangdik@who.int; 8Global Outbreak Alert and Response Network (GOARN), World Health Organization, SEARO, New Delhi 110011, India; delriov@who.int; 9Research and Training in Tropical Diseases (TDR), World Health Organization, Geneva 1211, Switzerland; zachariahr@who.int

**Keywords:** SORT IT, operational research, universal healthcare, SDGs, Bhutan, emergency ambulance performances

## Abstract

**Background:** An efficient ambulance service is a vital component of emergency medical services. We determined the emergency ambulance response and transport times and ambulance exit outcomes in Bhutan. **Methods:** A cross-sectional study involving real-time monitoring of emergency ambulance deployments managed by a central toll-free (112) hotline (20 October 2021 to 20 January 2022) was carried out. **Results:** Of 5092 ambulance deployments, 4291 (84%) were inter-facility transfers, and 801 (16%) were for emergencies. Of the latter, 703 (88%) were for non-pregnancy-related emergencies (i.e., medical, surgical, and accidents), while 98 (12%) were for pregnancy-related emergencies. The median ambulance response and patient transport times were 42 (IQR 3–271) and 41 (IQR 2–272) minutes, respectively. The median round-trip distance travelled by ambulances was 18 km (range 1–186 km). For ambulance exit outcomes that were pregnancy-related (*n* = 98), 89 (91%) reached the health facility successfully, 8 delivered prior to ambulance arrival at the scene or in the ambulance during transport, and 1 had no outcome record. For the remaining 703 non-pregnancy deployments, 29 (4.1%) deployments were deemed not required or refusals, and 656 (93.3%) reached the health facility successfully; 16 (2.3%) died before the ambulance’s arrival at the scene, and 2 (0.3%) were not recorded. **Conclusions:** This first countrywide real-time operational research showed acceptable ambulance exit outcomes. Improving ambulance response and transport times might reduce morbidities and mortalities further.

## 1. Introduction

Bhutan is a mountainous country, where road travel is challenging due to difficult terrain and winding roads. The public transport network is weak, and in rural areas, taxis and private cars are expensive and not widely available. In such a setting, an efficient ambulance service is a vital component of the emergency medical service system. This offers a first point of contact for pre-hospital care and timely evacuation of a patient from the incident site to a receiving health facility. Two studies from Ethiopia showed that ambulances are highly cost-effective in reducing mortality and improving patient outcomes [1,2]. A systematic review by Henry J. et al. showed that, where there is good pre-hospital care, there can be up to 25% reduced risk of death from trauma [3].

Ambulance availability, response time, competence of the ambulance team, and clarity of ambulance calls are all factors that can determine the effectiveness of ambulance services [2,4,5,6,7]. The emergency medical services division (EMSD) of the Bhutan Ministry of Health (MoH) has made efforts to improve the ambulance service network in the country. The number of equipped ambulances increased from 80 in 2011 to 98 in 2016, and reached 129 in 2022. Each ambulance deployment is accompanied by a qualified emergency medical responder (EMR), and all ambulance services are provided free-of-charge. An emergency toll-free (112) hotline was also established in the year 2011 to coordinate countrywide ambulance deployments. Since the beginning of the COVID-19 outbreak, the ambulances have also been provided with personal protective equipment (PPE), and all EMRs and ambulance drivers have been trained in PPE donning and doffing techniques.

The performance of ambulance services for improving emergency medical care remains largely unevaluated in low- and middle-income countries, including in Bhutan [8]. Anecdotal observations by family members, healthcare professionals, and the media have suggested delays in ambulance deployment in Bhutan. In high-income countries, the recommended benchmark response time for emergency ambulance services is under eight minutes [9]. No such standards exist for low- and middle-income countries. A study from Ethiopia reported an ambulance response time (time from call to arrival at the scene) of 10 min, and a transport time (time from reaching the emergency scene to arriving at a health facility) of 17 min [10]. Reductions in overall response time translate into increased probability of survival [4,11]. Other studies have shown that ambulance location and the use of satellite deployment decrease ambulance response times [5,6,7].

Knowledge of ambulance response times, the characteristics of individuals benefiting from the ambulance services (utilization), and their outcomes would be useful to assess the quality of care provided by the ambulance team. Such information would help us to better understand what works and what does not work, and trigger improvements. We aimed to describe the emergency ambulance network, its performance, and its outcomes in Bhutan. Our specific objectives were to determine (a) caller details and weekly trends of ambulance deployments, (b) ambulance response and patient transport times, (c) morbidities, and (d) ambulance exit outcomes.

## 2. Materials and Methods

### 2.1. Study Design

A cross-sectional study involving real-time monitoring data of emergency ambulance deployments from the national road ambulance network.

### 2.2. Settings

#### 2.2.1. General Setting

Bhutan is a country that lies on the southern slopes of the eastern Himalayas, between China and India. It has steep and high mountains crisscrossed by swift rivers that form deep valleys. Altitude ranges from 200 m in the southern foothills to more than 7000 m in the highlands. There is great geographical diversity, combined with diverse climatic conditions, including heavy rains during monsoon season and snowfalls in winter, which can disrupt road transport. The largest city in Bhutan is the capital, Thimphu. The country has a population of about 700,000, and is divided into 20 districts.

The health system is tiered. The country has one national referral hospital, two regional referral hospitals, 46 general hospitals, and 186 primary health centers. Furthermore, basic medical care and mother-and-child health services are routinely provided through several outreach clinics. Individuals from lower tiers of the health system may be referred to higher-level health facilities, based on clinical need.

#### 2.2.2. Specific Setting

##### The Health Help Centre (112 Hotline) and Ambulance Network

The 112 call center receives calls for general medical advice, as well as for ambulance services. Medical care, including all ambulance services, is provided free-of-charge, in line with the national ambulance services guidelines [12]. There are minimum standards for staffing, equipment, and on-board medication for ambulances [12].

The call center, based in Thimphu, coordinates all ambulance requests and deployments, and the service is available round the clock (24 h a day, all days of the week). The center is manned by 23 trained health workers, including nurses and health assistants, with additional training in medical emergencies. There are five call handlers at any moment in time. Ambulances are equipped with a global positioning system (GPS) and a mobile phone. This allows the 112 center to be in constant communication with ambulances and track their movements. This system allows prompt deployment of an ambulance(s) that is closest to the emergency scene.

Ambulance services can be utilized by anyone for medical illness, obstetric emergencies, accidental trauma, and other emergencies, such as road traffic accidents. The call center also coordinates inter-health-facility patient transfers, which may be planned or urgent.

All ambulances are stationed at health facilities, and ambulance allocation is determined by the number of hospital beds [12]. In emergencies, the ambulance team usually comprises a driver and an EMR, and this team can be augmented depending on the nature of the emergency. Each health facility is provided with a fixed budget per ambulance per year, to cover all maintenance and fuel costs. When this amount is used up, the ambulance may be grounded, but in most cases this does not happen, due to the allocation of supplementary budget when justifiable. Ambulance drivers function round the clock according to a schedule on a duty roster. The minimum number of ambulances allocated in relation to the number of hospital beds is shown in Table 1.

Figure 1 shows the distribution of different levels of health facilities and the number of ambulances allocated to each health facility at the time of the study. When an ambulance in a particular health facility is grounded due to mechanical failure or surrendered for disposal, mechanisms are in place to mobilize new or substitute ambulances to meet the requirements and provide uninterrupted services.

##### Study Population and Period

The study included all emergency ambulance deployments (those emergency deployments outside of the health facilities) following hotline calls (112) in Bhutan for a duration of three months between 20 October 2021 and 20 January 2022. Inter-health-facility patient transfers were excluded. The three-month period reflects the implementation of real-time operational research, whereby short periods of analysis are used to continuously inform the decision-making process for health system strengthening during public health emergencies.

### 2.3. Data Collection, Analysis, and Statistics

Data variables included call identity number, date and time of call, caller details in terms of the relationship between the caller and the patient, time of ambulance assignment, time ambulance arrived at the scene, time ambulance arrived at the health facility, total distance travelled, patient characteristics, morbidity, and standardized ambulance exit outcomes. The sources of data were the call center register and the ambulance trip sheets (Appendix A).

Data were double-entered on a weekly basis by staff working in the 112 call center into EpiData Analysis version 2.2.2.186 (EpiData Association, Odense, Denmark), and we used Stata v13 (Stata Corporation, College Station, TX, USA) for further analysis. Trends in ambulance deployments were presented weekly, and numbers of deployments per district were standardized per 10,000 population, using the district population estimated by the Bhutan Population and Housing Census of 2017. Ambulance response time was calculated as the difference between the time of a 112 call and the time the ambulance reached the emergency scene. Patient transport time was calculated as the difference between the time the ambulance reached the scene and the time of reaching the health facility. Continuous data (i.e., age, ambulance transport and response times) were assessed for normality using the Shapiro–Wilk test. Depending on the shape of the data curve, means or medians were presented. Distances travelled by ambulances were for round trips (from start location to return). Reasons for ambulance calls and ambulance exit outcomes were standardized separately by the research team for pregnancy-related and medical-related outcomes, as detailed in Table 2.

## 3. Results

During the three-month study period, the 112 call center received a total of 16,207 calls for various reasons, of which 5092 resulted in ambulance deployments. Of the latter, 4291 (84%) were for inter-facility transfers, and 801 (16%) for emergency deployments at locations outside the health facilities. The summary of all ambulance deployments and outcomes is depicted in Figure 2. This study concerns only the emergency ambulance deployments.

### 3.1. Caller Details and Trend of Ambulance Deployments

The caller details for the 801 ambulance deployments were as follows: 492 (62%) were parents and relatives, 55 (7%) were friends, 64 (8%) were neighbors, 18 (2%) were office colleagues, and 101 (13%) were passers-by. Caller details were not recorded for 71 (9%) ambulance deployments. Figure 3 shows weekly trends of ambulance deployments.

The average number of ambulance deployments was 57 per week (range: 35–115) and 8 per day (1–15). Deployments of ambulances per 10,000 population and by district are presented in Table 3. The highest numbers of emergency deployments were in the weeks of 1–7 December 2021 and 12–18 January 2022. Following standardization, the top five districts for ambulance deployments were Punaka, Sarpang, Dagana, Trashigang, and Bumthang.

### 3.2. Ambulance Response Time and Patient Transport Time

Table 4 shows ambulance response times and patient transport times. The overall median ambulance response time was 42 min, and median patient transport time from the scene to the health facility was 41 min. The median ambulance response time for those who died was 52 min (range: 9–145). The median total distance travelled by ambulances was 18 km (range: 1–186 km).

### 3.3. Patient Characteristics and Morbidities

The characteristics of the patients are shown in Table 5. Of the 801 patients who availed ambulance services, 424 (53%) were female, and the median age was 38 years (range: 1–102 years).

Patient morbidities included medical conditions (*n* = 531, 66%), pregnancy (*n* = 98, 12%), accidental trauma (*n* = 82, 10%), road traffic accidents (*n* = 29, 4%), and surgical conditions (*n* = 15, 2%). Patient morbidity was not available for 46 deployments.

### 3.4. Ambulance Exit Outcomes

Ambulance exit outcomes were categorized into non-pregnancy-related (*n* = 703) and pregnancy-related (*n* = 98). In 29 (4.1%) of 703 ambulance deployments for non-pregnancy-related (i.e., medical, surgical conditions, and accidents) emergencies, ambulance transfer was not required, or was refused by the patient despite them calling for the ambulance initially. Of the 29 refusals, 6 travelled by their private cars despite calling for an ambulance, 16 refused to go to hospital after receiving treatment at the scene/home, 1 did not respond to a call after the ambulance reached the scene, and no reason was provided for the remaining 6. In 656 (93.3%) cases, the patient was safely handed over to the health facility. There were 16 deaths, of which 14 occurred before the ambulance arrived at the emergency scene, while two were en route to the health facility.

Of deployments that were pregnancy-related (*n* = 98), 89 (91%) were safely handed over to the health facility, while 8 either delivered prior to the ambulance’s arrival at the scene or delivered in the ambulance. These details are shown in Figure 2.

## 4. Discussion

This first countrywide study of the emergency road ambulance network in Bhutan showed that median ambulance response times and patient transport times were under one hour, with over 90% of emergencies being transported safely to health facilities. This is encouraging, considering the mountainous terrain, poor roads, and icy conditions that are prevalent in the country.

These findings are of importance to public health, as an efficient ambulance network is vital for improving access to health facilities, achieving universal health coverage, and protecting the population from public health emergencies, in line with achieving the WHO’s Triple Billion targets [13]. In this study, weekly ambulance deployments fluctuated remarkably, but no definite cause could be identified. Absence of deployment data from Tsirang District was due to the non-submission of ambulance trip sheets by the ambulance team, rather than the lack of ambulance services in the district.

This study’s strengths are that it was a countrywide assessment, data collection was carried out by trained staff, and the subject per se was an identified national operational research priority, as set out in the Ministry of Health’s emergency medical services performance indicators [12]. Furthermore, this is an example of operational research being conducted in real time, for real-time decision making, for real-time action. Such an approach is important during public health emergencies, such the COVID-19 pandemic, which constantly disrupts routine health services. We also adhered to the STROBE guidelines for reporting of observational studies in epidemiology [14].

One of the main limitations of our study is that our dataset included a three-month preliminary period, and we have thus not captured temporal and seasonal patterns, which are highly variable in Bhutan over the four seasons. However, this was deliberate, as this study is real-time operational research, which involves an approach of conducting frequent data audits for generating timely evidence that can show “what works and what does not work” to inform corrective measures. We had 6% missing data on morbidities which, along with the finding of missing data on a number of variables, highlights the need for greater programmatic vigilance. Future research should assess ambulance movements—emergency or otherwise—affected by lockdowns and other effects of the COVID-19 pandemic and seasonal patterns.

This study has a number of policy and practice implications. Although median ambulance response time and patient transport time amounted to roughly 41–42 min, we have no local standards for comparison to determine whether this is within acceptable limits. For obstetric emergencies, the recommended time to reach emergency services in health facilities has been estimated at two hours [15]. As distances, road conditions, and geographic terrains in Bhutan are variable, and can independently affect ambulance response and transport times, setting such standards is difficult. A study from Ethiopia [10] reported ambulance time (response time + transport time) to be 27 min, which is much lower than ours, but the geographic terrain is considerably different. In the absence of a standard for Bhutan, it would seem reasonable to use ambulance exit outcomes as a proxy for performance. The ambulance exit outcomes in this study were reasonably acceptable, with more than 90% favorable outcomes. However, improving ambulance assignment and dispatch time may be an effective intervention to reduce delays, and is perhaps more amenable to change than travel time.

In 29 (4.1%) deployments, ambulance transport was refused after it was called to the scene/home. Deeper analysis of these refusals revealed that six of them took their private cars to the hospital despite calling the ambulance for emergency. These could have resulted from delays in ambulance arrival or patients feeling uncomfortable traveling by ambulance. In addition, 16 patients refused to go to the health facilities after the ambulance’s arrival and receiving onsite medical treatment by the ambulance team. This could be a good indication that, where possible and feasible, onsite treatment should be provided and strengthened to reduce unnecessary transport and admission of patients to hospital. If EMRs are trained in the provision of broader basic medical, surgical, and obstetric care beyond accidents and trauma, the proportion of patients admitted to hospitals may reduce, thereby avoiding hospital overload, bed shortages, and inconveniences to the people.

There were 16 deaths, 14 of which happened prior to the arrival of the ambulance, while 2 happened enroute to the health facility. Five of these deaths were listed as “found dead”, one was listed as cardiac arrest, one assault causing death, one immediate death at accident site from road traffic accident, one death due to forest fire, one elderly, bedridden patient who died before an ambulance arrived, and the rest (six) were not detailed. The median ambulance response time was longer (52 min) in those who died compared to those who were transported to health facilities successfully (42 min). These findings indicate that improving ambulance times might contribute to preventing deaths. Although these are a clear indication of the severity of illness, indicating the need to shorten the ambulance response time, it may be arguable that prompt ambulance services could reduce such incidences, due to the nature of the deaths described. It may do more good to analyze the causes of sudden deaths and advocate preventive measures rather than improving ambulance services.

A possible way of improving the response time is to improve the location and distribution of ambulances [5,6,7]. Furthermore, for life-threatening emergencies, it would be logical to consider the use of helicopter ambulances, which are also available and provided free for medically indicated cases. This would considerably reduce the time lost through long and difficult road travel. The Bhutan Emergency Aeromedical Response (BEAR) has two helicopters at its disposal, but optimizing their use would depend on the clarity of calls for identification of those in a life-threatening situation. Since the overall estimated effect of reduced ambulance response time has shown improved survival, this is a measure to be considered for reducing deaths [4,11]. Information education and communication on the criteria for helicopter deployment and training of ambulance management staff would need to be an integral aspect of such a strategy. With this learning, different remedies—including the implementation of plan–do–study–act (PDSA) rapid improvement cycles—with the available resources should be engaged to improve ambulance response times.

Eight pregnant women delivered prior to reaching the intended health facilities. We do not have the details on whether these were high-risk pregnancies, and this would be an aspect to consider capturing in the routine monitoring and evaluation system. Obstetric emergencies need rapid response in order to save the lives of both the mother and the baby, and this is an area to focus on for further improvement.

Finally, only 16% of all ambulance transfers were emergency-related, and the remainder were inter-hospital transfers. This indicates that improving specialist medical services in more hospitals across the country would reduce ambulance requirements. To better understand the overall role played by the ambulances in providing access to emergency services, one would need to gather data on emergencies that arrived at health facilities using all available means of transport, and the proportional contribution made by the ambulances. This is an area that merits further investigation.

## 5. Conclusions

This first countrywide study from Bhutan involving real-time operational research showed acceptable ambulance exit outcomes, and highlighted a number of areas meriting improvements. In view of making improvements, the findings will be discussed with decision makers, and we will continue to use real time operational research to assess the impacts of future interventions. More elaborate studies focused on ambulance deployments and outcomes specific to accidents and trauma, obstetrics, and other medical emergencies may reveal in-depth needs for improvement strategies.

## Figures and Tables

**Figure 1 tropicalmed-07-00087-f001:**
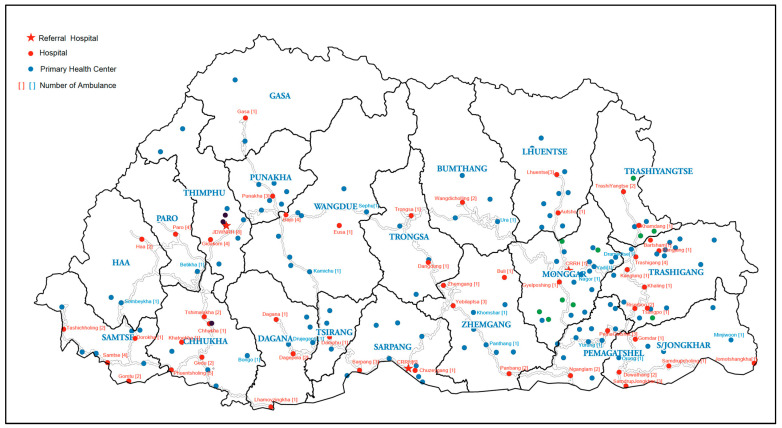
Locations of health facilities and numbers of ambulances at each facility in Bhutan.

**Figure 2 tropicalmed-07-00087-f002:**
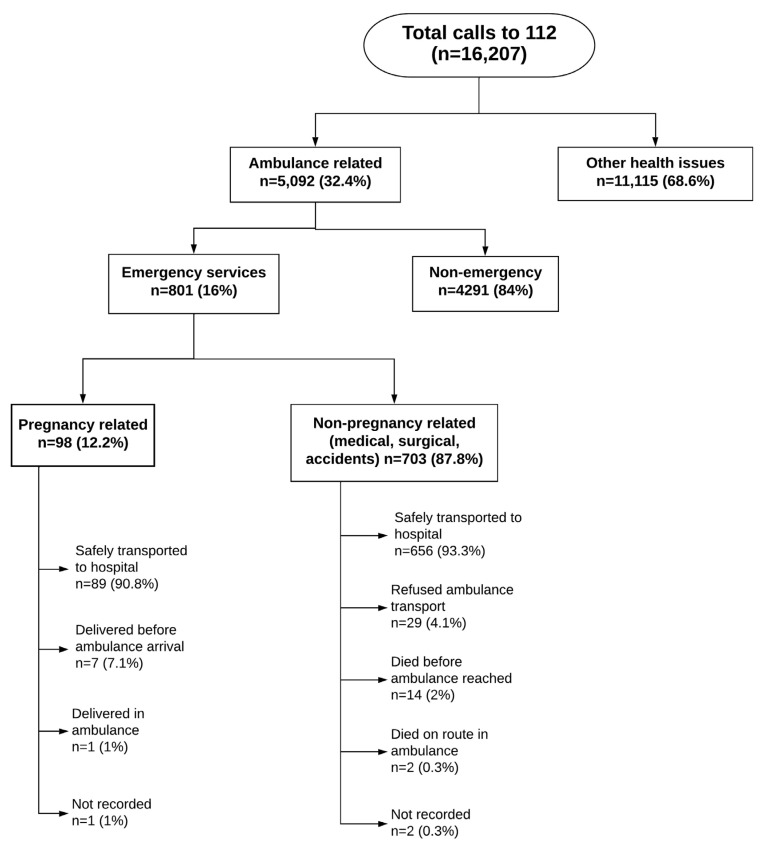
Flowchart showing total ambulance deployments and emergency deployment outcomes.

**Figure 3 tropicalmed-07-00087-f003:**
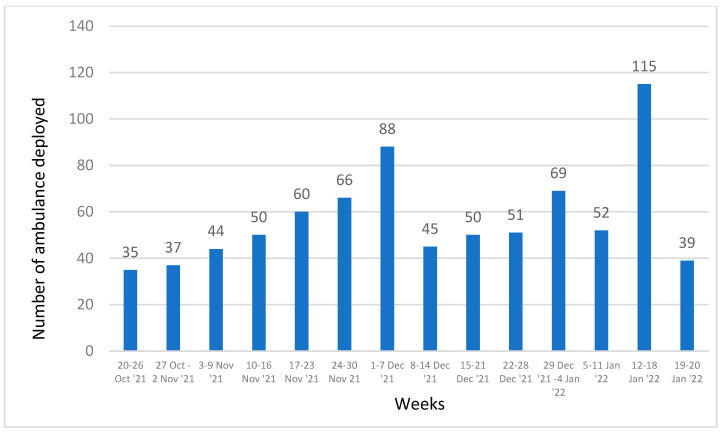
Weekly trends of ambulance deployments in Bhutan (20 October 2021 to 20 January 2022).

**Table 1 tropicalmed-07-00087-t001:** Numbers of ambulances allocated by health facility level in Bhutan.

Level of Health Facility	Number of Hospital Beds	Number of Allocated Ambulances
National referral hospital	350	6
Regional referral hospital	150	5
General/larger district hospital	40–60	3
Smaller district hospital	20–39	2
Sub-district hospital	10	1

**Table 2 tropicalmed-07-00087-t002:** Reasons for ambulance calls and ambulance exit outcomes.

Reasons for Call	Ambulance Exit Outcomes
Medical case Surgical case Accident trauma Road traffic accident Pregnancy related Others (specify)	**Non-pregnancy-related outcomes:** Alive and transported safely Died before ambulance reached scene Died en route to hospital Died on arrival at hospital Ambulance transfers not required/refused Not applicable/not recorded
	**Pregnancy-related outcomes:** Safely reached hospital Delivered before ambulance reached scene Delivered in the ambulance Ambulance transfers not required/refused Not applicable/not recorded

**Table 3 tropicalmed-07-00087-t003:** Total ambulance deployments standardized by district population in Bhutan (20 October 2021 to 20 January 2022).

District	District Population	Total Ambulance Deployment	Standardized Deployment ^1^
Punakha	28,842	69	23.9
Sarpang	46,571	88	18.9
Dagana	24,648	43	17.4
Trashigang	44,984	72	16.0
Bumthang	17,471	26	14.9
TrashiYangtse	16,929	25	14.8
Zhemgang	17,648	25	14.2
Monggar	36,526	47	12.9
Haa	14,374	18	12.5
Chhukha	71,473	74	10.4
Lhuentse	13,852	12	8.7
Paro	46,717	39	8.3
SamdrupJongkhar	34,949	29	8.3
Thimphu	139,726	116	8.3
Trongsa	22,763	16	7.0
WangduePhodrang	46,157	31	6.7
Samtse	61,353	39	6.4
Pema Gatshel	23,247	14	6.0
Gasa	3910	2	5.1
Tsirang ^2^	22,186	0	0.0

^1^ Per 10,000 population; ^2^ there were no ambulance trip sheets received from this district.

**Table 4 tropicalmed-07-00087-t004:** Ambulance response times (in minutes) to reach the emergency scene, and patient transport times to the health facility, in Bhutan (20 October 2021 to 20 January 2022).

	Median	IQR	Range
Time from call to ambulance being assigned	4	3	1–178
Time from being assigned to ambulance deployment	6	8	1–186
Time from deployment to reaching the scene	30	44	1–271
**Ambulance response time ^1^**	**42**	50.5	3–271
**Patient transport time ^2^**	**41**	49	2–272

**^1^** Calculated from time the call was received to reaching the scene of the emergency; **^2^** time for transporting the patient from the emergency scene to the hospital.

**Table 5 tropicalmed-07-00087-t005:** Characteristics of patients who used ambulances for emergency deployment in Bhutan (20 October 2021 to 20 January 2022).

Characteristics	Category	Number	(%)
		801	
Age category (years)			
	Under 5 (≤5)	39	(4.9)
	6–18	57	(7.1)
	19–30	194	(24.2)
	31–60	317	(39.6)
	≥61	186	(23.2)
	Not recorded	8	(1.0)
Sex	Male	365	(45.6)
	Female	424	(52.9)
	Not recorded	12	(1.5)
Occupation	Unemployed	94	(11.7)
	Employed	91	(11.4)
	Housewife	98	(12.2)
	Student	48	(6.0)
	Farmer	305	(38.1)
	Military	3	(0.4)
	Not recorded	162	(20.2)

## Data Availability

The data has been presented in this study and raw data is available on request from the corresponding author.

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
