# Peer review of "Performance of an Emergency Road Ambulance Service in Bhutan: Response Time, Utilization, and Outcomes"

_tropicalmed, 2022, doi:10.3390/tropicalmed7060087_

Round 1

Reviewer 1 Report

I enjoyed reading this paper and it is interesting to see how ambulance services operate in Bhutan.  The results are very descriptive but informative.  As the survey was conducted during the Covid-19 pandemic it would have been useful to assess the likely impact on the results of this study.  The references are rather dated but are mostly applicable.  The paper is well written in good English - just one typo line 305 'indicates' rather than 'indicate'.

Reviewer 2 Report

Many thanks for an informative and important publication. Glad to see successful outputs from the SORT-IT programme and hope the capacity built and operational insight developed will help strengthen pre-hospital care in Bhutan.

The draft paper is informative. The setting section is well described and provides a useful overview of the health system including the ambulance system.

The methods section is mostly clear and well described. The results are interesting, however improving the sequence of result reporting will improve the flow and facilitate comprehension.

The discussion could be improved with clearer links from results to discussion topics. I provide comments and suggestions below.

I advise the authors to discuss seasonality and whether the three month results are generalizable to the rest of the year (the authors mention Monsoon season). Work in Sierra Leone (Caviglia et al),a s one example, demonstrates longer response times during rainy season.

Given the fluctuations in utilization demonstrated by week and district, it would be helpful to have more analysis and discussion focused on this. Are there high/low performing districts in terms of response time? This analysis could help target improvements.

Improving assignment and dispatch time is a possible intervention that the results highlight, this has been successfully reduced in other countries and is more amenable to change than travel time.

Other comments

Line 25 “ambulance exit outcomes” is there a better term for this?

Line 30 Please present interquartile range with median

Line 50-52 “If appropriately used, ambulances are highly cost-effective in reducing mortality and improving patient outcomes[1,2”]  The reviewer’s opinion is that the evidence base to state whether ambulance systems are highly cost-effective is very small. "One study of xx patients in one country has shown ambulances to be highly cost-effective" , may be more appropriate phrasing.

Line 55: Efficiency or effectiveness?

Line 70: recommended benchmark for “emergency cases?”

Line 111 “available around the clock” Please state horus and days the service is available for.

Figure 1 is too small for the reviewer to interpret, what is PHC? It may be helpful to show the road system alongside the ambulance locations.

Line 148: Who entered the data? Was the data routinely captured by the ambulance call centre, then re-entered in EpiData?  Please clarify.

Line 158: Where was district population obtained from?

Line 164: spelling, median? Please present mean with standard deviation and medians with IQR.

Line 166: Who standardized the reason? The call centre operator? The research team?

Line 172: What is an emergency deployment? Please define. Do the authors mean a location which is not a health facility? Or are inter-facility transfers not classified as emergencies?

Line 196: It is helpful to see the difference in response time for those that died, yet we have not been told how many patients died yet. Was the response time also elevated? Was this significantly different to those that survived?

Table 2 is helpful:  Other publications use the terms, Dispatch time, travel time, time on scene, travel time. And the overarching time from emergency call to patient arrival is “response time”.  Surprising that despite the extra assignment time and dispatch time that the Ambulance response time and patient transport time are almost the same. Would be interesting to hear the authors views on this in the discussion? Is this “on scene time”?

Figure 3 is very helpful, I suggest this is presented  earlier in the results section.

Line 258 for obstetric care there is a two hour minimum standard. (WHO. Monitoring emergency obstetric care: a Handbook, WHO,
UNICEF, UNFPA and AMDD. World Health Organization, 2009.)

And Lancet commission of global surgery.

Line 249: Please comment on seasonality? Monsoon, dry season? How may this have affected the results?

Line 281: What is a “spot death”?

Line 289: Another possible option to improve the response time is to try and improve your dispatch time: “Time from call to ambulance being assigned” and “Time from being assigned to ambulance deployment”. If the reviewer has interpreted correctly, you are losing a median of 12 minutes here. There are a variety of simple pragmatic interventions to reduce dispatch time which can be improved via QI/PDSA cycles.

Discussion

Please add comment about weekly trends, reported in figure 2, whether this is a data capture issue or real fluctuation in utilization?

Some comment about utilization, why no utilization in Tsirang district for example?

Thank you for the informative work.
